# Mixed-methods feasibility outcomes for a novel ACT-based video game 'ACTing Minds' to support mental health

Tom C Gordon,[1,2] Andrew H Kemp ,[1] Darren J Edwards [2]

¹School of Psychology, Swansea University, Swansea, UK
²Department of Public Health, Swansea University, Swansea, UK

**Correspondence to**
Dr Darren J Edwards;
D.J.Edwards@swansea.ac.uk

## ABSTRACT

**Objectives** To determine the feasibility and acceptability of 'ACTing Minds', a novel single-player adventure video game based on acceptance and commitment therapy (ACT).

**Design** A single-arm, mixed-methods repeated measures feasibility study.

**Setting** Intervention and questionnaires were completed at home by participants. Semistructured interviews were also conducted at home via the Zoom platform.

**Participants** Thirty-six participants were recruited into the study, 29 completed all phases of the feasibility design. Eligibility criteria required participants to be over the age of 18 and self-reporting experiencing ongoing depression, anxiety or stress.

**Intervention** Participants completed a single session of the 'ACTing Minds' video game, lasting approximately 1 hour, designed to educate users on key principles from ACT.

**Primary outcome measures** Participant recruitment and retention, questionnaire completion, long-term intervention adherence and acceptability of the intervention. Reflexive thematic analysis was conducted on semistructured interviews run immediately postintervention and 3 weeks later.

**Secondary outcome measures** Measures of depression, anxiety, stress, psychological flexibility, social connectedness and well-being were assessed at baseline, immediately following intervention completion, and after a 3-week follow-up period. We used a standardised battery of questionnaires.

**Primary results** Twenty-nine participants completed the study. A reflexive thematic analysis indicated that participants responded positively to the intervention and the study at all stages. Themes reflect participants' desire for an engaging therapeutic experience, use of game for exploring emotions, as well as their perspectives on how they had applied their learning to the real world.

**Secondary results** Quantitative results indicated small to large effect sizes associated with decreases in depression (ηp2 = 0.011), anxiety (ηp2 = 0.096) and stress (ηp2 = 0.108), and increases in psychological flexibility (ηp2 = 0.060), social connectedness (ηp2 = 0.021), well-being (ηp2 = 0.011) and participation in usual activities (ηp2 = .307).

**Conclusions** Implementation of the 'ACTing Minds' intervention is warranted, based on both qualitative and quantitative outcomes.

## STRENGTHS AND LIMITATIONS OF THIS STUDY

⇒ Mixed methods approach, combining thematic analysis of interviews and quantitative questionnaires.

⇒ Collection of quantitative data at three time points and qualitative at two time points, allowing the process of change and identification of patterns to be examined.

⇒ Remote data collection due to COVID-19 restrictions meant that participants could not be directly observed while completing the intervention. We were also unable to record planned psychophysiological measurements of well-being such as heart rate variability.

⇒ Reliance on self-report measures introduces the potential for bias.

**Trial registration number** NCT04566042 ClinicalTrials.gov

## INTRODUCTION

The global prevalence of common mental disorders and a lack of available resources for the identification and treatment of those with such conditions underpin an increasing burden on society.[1] The Global Burden of Disease study conducted in 2017 reported a UK prevalence rate of 4.12% for depression and 4.65% for anxiety disorders.[2] Since this estimation, events such as the COVID-19 pandemic and the increasing threat of climate crises have had a substantial impact on societal well-being. A meta-analysis including 14 studies (n=46158) found that 32% of adults in the UK experienced moderate to severe depressive symptoms in 2022, and 31% of adults reported high levels of anxiety,[3] indicating a societal increase of 27.88% for depression and 26.35% increase for anxiety between 2017 and 2022. These findings suggest the need for a transition towards population-wide strategies aimed at fostering psychological resilience on a broader scale.

To positively impact societal well-being, contemporary interventions must be affordable and widely accessible. Presently, the

demand for mental health services far exceeds the available human resources required to meet this need. A study conducted for the Centre for Mental Health estimated that services cost the UK economy approximately £105 billion per year in 2020, 4.8% of the UK's annual GDP.[4] Despite substantial funding of £34 billion to public mental health support and services, the prevalence of psychological disorders is high and only 33% of adults with depression and anxiety receive treatment in England,[5] highlighting an urgent need for innovation.

There are numerous barriers to accessing psychological interventions, including a shortage of therapists, long waiting times and societal stigma of accessing psychological treatment.[6] A potential solution to these issues might be found in digital health interventions (DHIs). We live in an age of heavy digital media consumption, especially in the West where at least 90% of UK adults use the internet regularly.[7] We also know that during the COVID-19 pandemic, there were significant increases in online video consumption, social media usage, remote work, online news consumption and video gaming.[8] COVID-19 contributed to significant social isolation and further disconnection from nature, further contributing to increases in mental health conditions.[9] We argue that there is an opportunity to leverage technological advancements and the growing use of technology to develop psychoeducational tools necessary to support mental health at scale.

DHIs have already been used in a variety of contexts for promoting well-being, from delivering healthcare and education to personalised diet and fitness plans. Mobile apps and online platforms offer guided meditation, breathing exercises, sleep tracking and relaxation programmes. Such applications might aim to enhance overall well-being, reduce stress, improve sleep quality or cultivate mindfulness practices.[10] We suggest that effectively addressing well-being at a population level should involve the development of DHIs that consider acceptability, feasibility, and widespread appeal.

Compared with alternative forms of media, video game DHIs offer several advantages. By design, they are interactive, applying behavioural principles for controlling and modifying behaviour,[11] making them uniquely captivating. In the UK, the COVID-19 pandemic led to a substantial increase in the number of people playing video games, with males increasing their use from 46% to 63% and females increasing their use from 32% to 56% in 2022.[12] Innovations in the use of video games for treating mental health issues have wide potential applications, potentially offering a platform for individuals to explore their ongoing relationship with their emotions in a supportive environment.

In theory, by practising skills derived from psychological therapies (such as ACT) within the game context, individuals can transfer these skills to real-life situations to improve their overall quality of life. Certain games specifically designed for therapeutic purposes, such as 'SPARX' for depression[13] or 'Elude' for anxiety,[14] guide players through interactive challenges and cognitive exercises for developing emotional regulation skills. Video games are also being used in clinical settings to promote well-being outcomes. One game designed for this purpose, 'Dojo',[15] aims to treat anxiety by training users in breathing techniques, muscle relaxation, positive thinking and guided imagery, using heart-rate variability (HRV) biofeedback. However, when compared against a standard commercial game 'Rayman 2' (control condition), a full pre-post randomised controlled trial (RCT) (n=138) found that playing either game significantly reduced participant anxiety at a 3-month follow-up, and there were no significant differences between these two games at reducing anxiety for this time period.[15] Reasoning for this might be that 'Dojo' failed to develop the psychoeducational skills in the participants that it aimed to impart, or that both games only reduced anxiety by distracting (as a form of avoidance) participants from anxiety-provoking thoughts.[16] The researchers concluded that 'Dojo' had crucial design issues that needed to be addressed including a lack of clear theoretical and therapeutic frameworks, and that research assessing the real-world effectiveness of video games in the treatment of mental health issues should require an appropriate methodology for understanding the underlying causes of improvement.

A study aiming to explore the well-being effects of playing video games on gamers during the COVID-19 pandemic (n=781) found that time spent playing had significantly increased in 71% of participants, and 58% of participants reported that playing games had positively impacted their well-being.[17] The researchers conducted an online survey including both closed and open-ended questions, then conducted a thematic analysis to identify the causes of improvement. Themes of escape, cognitive stimulation, stress relief, agency and socialisation were most associated with feelings that playing video games had increased well-being. The development of an effective DHI video game should consider such factors while also building on strong theoretical and therapeutic foundations that facilitate the uptake of such tools.

The 'ACTing Minds' video game, developed in line with our intervention protocol,[16] was designed to be a comprehensive transdiagnostic intervention that will integrate lessons from acceptance and commitment therapy (ACT).[18] In contrast to prior mentioned DHIs, commonly rooted in the medical model and second-wave approaches, ACT as a third-wave behavioural therapy focuses on promoting psychological flexibility rather than the elimination of disorder symptoms.[19] More specifically, ACT aims to promote psychological flexibility through six core processes of change.[20] These are[1] present moment awareness: the practice of being in the here and now[2]; acceptance: the practice of being open to the range of human emotional experience, as opposed to experiential avoidance[3]; cognitive defusion: the act of recognising the self as separate from thoughts, and not interpreting them literally[4]; values: identifying ones' personal values in contrast to perceived expectations, of which drive us

toward self-direction and purpose[5]; action: a commitment to ones' values, facilitating the development of competence through the act of continual practice of alignment with values[21]; and[6] self as context: developing an awareness of self that is more than a conceptualised sense of self, one that is flexible and facilitates a sense of connection with others.

Though ACT clinical practice does not focus primarily on reducing mental health symptoms, many studies have indicated that when the individual works towards greater psychological flexibility, many mental health symptoms, and destructive behaviours such as anxiety, depression, stress, pain and substance misuse, tend to reduce with clinically acceptable small to high effect sizes. This was, for example, identified within a review of 20 meta-analyses, involving 133 studies (n=12 477) that found that ACT was efficacious for treating these disorders.[22] The results also showed that ACT was generally superior to most active intervention conditions (excluding CBT), treatment as usual, and inactive controls.

As such, an ACT-based DHI video game could allow for both greater psychological flexibility as well as a reduction in common mental health issues such as depression and anxiety. This is because developing explicit DHI psychoeducational transdiagnostic skills that promote present moment awareness, values orientation, commitment to action, openness, and acceptance of painful emotions, cognitive defusion, and a transcendental self also have a positive impact on mental health. As a consequence, our video game may have greater reach and impact than other video game DHIs that are not based on third-wave psychotherapy, such as 'Dojo', which primarily aims to teach skills for limited emotional regulation and symptom reduction such as avoidance.

The ACT framework has already been adapted to a variety of accessible mediums, including self-help books and mobile phone applications.[23] Resources for the education and practice of ACT are freely available through the Association of Contextual Behavioural Science website (see https://contectualscience.org/). ACT-based mobile phone applications are shown to be effective in promoting psychological flexibility[24] and reducing smoking intake.[25] Considering this, we believe that choosing ACT as the basis for our game will allow us to harness the advantages of third-wave therapies as transdiagnostic therapeutic tools and integrate these with those of videogames, and if made well, will be engaging, educational, and capable of promoting well-being (psychological flexibility) at scale.

Based on our initial protocol[16] (see online supplemental file 1), the ACT-based video game called 'ACTing Mind' has been developed as a psychoeducational tool that teaches users the core processes of ACT through embedded learning. The goal of this research will be to determine the acceptability and feasibility of 'ACTing Minds' for promoting psychological flexibility as well as its clinical relevance for reducing mental distress as a secondary outcome. The game teaches skills based on the ACT principles of acceptance, defusion and values

identification. This is a feasibility study, following the Medical Research Council (MRC) framework,[26] laying the foundation for a full-scale RCT from which clinical effectiveness will be determined.

Several changes have been made to our originally published protocol;[16] it was initially stated that participants would complete five weekly 1-hour sessions where they would play through six parts of the 'ACTing Minds' video game, each one focusing on a different process of ACT. However, because of funding restrictions, 'ACTing Minds' has been compressed into a single game focusing on the ACT principles of acceptance, values identification and defusion. Therefore, in this feasibility study, participants will be required to complete a single 1-hour session of 'ACTing Minds'. This meant a significant change to the overall time to complete the study protocol. Originally, it was expected to take 3 months between baseline measurements and the final follow-up. Now, one-on-one semistructured interviews will be conducted immediately postintervention and after a 3-week follow-up. The research questions are as follows: Is the intervention acceptable and feasible for a full-scale RCT? Is there early evidence for effectiveness in reducing mental distress? Are there any changes in self-reported well-being measures following completion of the game? Are participants able to learn ACT principles and apply them in day-to-day life?

## METHODOLOGY
### Design
This is a single-arm, mixed-methods repeated measures study, designed to determine the feasibility and acceptability of an ACT-based video game called 'ACTing Minds' that has been designed for individuals reporting mild to moderate anxiety, depression and stress. Data were collected at baseline, immediately postintervention, and 3 weeks postintervention. Data collection was conducted between 1 November and 31 December 2022.

### Study setting
The study was conducted entirely online by participants, including the intervention (via a link to the mobile app; https://shorturl.at/iqFGI), quantitative assessment (via the Qualtrics platform) and qualitative interviews (via the Zoom platform). Strict recommendations were given to participants to ensure they were in a quiet room and without disruption for all study components.

### Participants
Thirty-six participants were recruited, 29 of which completed all phases of the study. Participants were recruited using purposive sampling methods, they were required to be at least 18 years of age, self-reporting ongoing experience of mild to moderate depression, anxiety or stress within their day-to-day life, and able to read, write and speak English. The sample size was justified on the basis of past research reporting the median numbers of participants recruited for similar feasibility

studies incorporating both quantitative and qualitative elements.[27] Advertisements were posted at Swansea University notice boards and on social media pages (Facebook mental health community groups). Participants were recruited between 1 October and 1 December 2022, they completed a consent form (see online supplemental file 2) after reading the study information sheet (see online supplemental file 3) and were given a debrief sheet (see online supplemental file 4) following completion of the study, each is included as supplementary materials.

### Primary outcome measures

Feasibility outcomes were determined using the MRC framework[26] and reported in line with CONSORT guidelines[28] (see online supplemental file 5). Feasibility measures included data relating to participant recruitment and retention including the number of participants willing to take part, and completing each stage of the study (ie, intervention, questionnaires, interviews and follow-up). Acceptability and efficacy of the intervention were assessed through thematic analysis of semistructured interviews conducted immediately postintervention and 3 weeks postintervention, which focused on participant experiences with 'ACTing Minds'. The first interview (see Open Science Framework for first interview protocol: https://osf.io/5fvjs) asked questions about[1] the acceptability of the intervention[2]; what they learnt from the intervention[3]; suggestions for further improvement[4]; whether there were any difficulties (barriers) in taking part[5]; aspects they liked and disliked; and[6] were there any adverse effects that they noticed while playing the game. This was followed by a second interview (see Open Science Framework for second interview protocol: https://osf.io/32epw) that was focused more on the real-world impact that ACTing Minds had on their lives and their experience over the 3-week period. Specifically, the second interview asked[1] about their retrospective experience in taking part in the ACTing Minds intervention[2]; how much they remember about the core ACT concepts[3]; were any aspects more memorable than others[4]; had they implemented any of the ACT concepts that they learnt while playing the game into their day to day lives[5]; had they found that any particular ACT concepts were more applicable to their everyday lives than others[6]; would they reuse the intervention; and[7] had they noticed any adverse effects in the 3 weeks since playing ACTing Minds.

### Secondary outcome measures

Questionnaires were distributed at three points in time (baseline, immediate postintervention and 3-month follow-up). A rule was created in Qualtrics requiring participants to complete every questionnaire item in order to finish the survey, which included the following questionnaires:

Depression Anxiety Stress Scales (DASS-21): a measure of general psychological distress with good construct validity (confirmatory factor analysis of 0.94) that can be broken down into subscales relating to stress, anxiety and depression. It has good internal reliability as measured through Cronbach's alpha coefficients, which are 0.88 for depression, 0.82 for anxiety, 0.90 for stress, and 0.93 for the total scale.[29]

Acceptance and Action Questionnaire-second version (AAQ-II): a questionnaire of seven items, assessing psychological inflexibility by gauging one's capacity to embrace and remain receptive to challenging thoughts and emotions, while also actively participating in meaningful actions despite their presence. A higher score indicates higher psychological inflexibility. The measure has good construct validity with a Cronbach's alpha coefficient of 0.84.[30]

Social Connectedness Scale (adapted from Russell's (1996) UCLA Loneliness Scale)[31]: this measure involves two questions[1]: "During social interactions, I feel 'in tune' with the person/s around me", and[2] "During social interactions, I feel close to the person/s". The Cronbach's alpha coefficients for these two items ranged from 0.80 to 0.98 (M=0.94, SD=0.03).[32]

Warwick-Edinburgh Mental Well-Being Scale (WEMWBS): a metric that emphasises the positive facets of mental health, aiming to evaluate overall psychological well-being. This measure has good internal consistency with a Cronbach's alpha coefficient of 0.89 (student sample) and 0.91 (general population sample).[33]

EuroQol Five Dimensions (EQ5D)[34]: a measure for health-related quality of life. There are five components within this measure which assess mobility, self-care, usual activities, pain, discomfort, anxiety, and a visual analogue scale for measuring current health status.

### Intervention

The intervention comprised of an ACT-based video game intervention called 'ACTing Minds', developed and designed by author DJE. Participants attended a single session lasting approximately 1 hour, during which they completed four in-game chapters.

The intervention teaches three core principles of ACT through embedded learning, meaning that the player should gain ACT-based skills while completing in-game objectives, without being directly taught those skills. These skills include 'Acceptance', 'Cognitive Defusion' and 'Values Identification'. Embedded learning refers to the incorporation of educational elements into the gameplay itself, in contrast to explicit lessons. In this context, it involves designing the game in a way that promotes psychological flexibility-oriented behaviours derived from ACT, such that the adoption values orientation, present moment awareness, openness to pain and cognitive defusion. In 'ACTing Minds,' an example of embedded learning is the 'Psychoflexameter' (see figure 1A), which serves as a gamified version of the Hexaflex (see figure 1B), a model used in ACT to illustrate both the theory and goals for clinical change.[35] Similar to the Hexaflex, the 'Psychoflexameter' showcases the six core processes of ACT. Initially introduced to players during the first ACT-oriented activity in the game, which

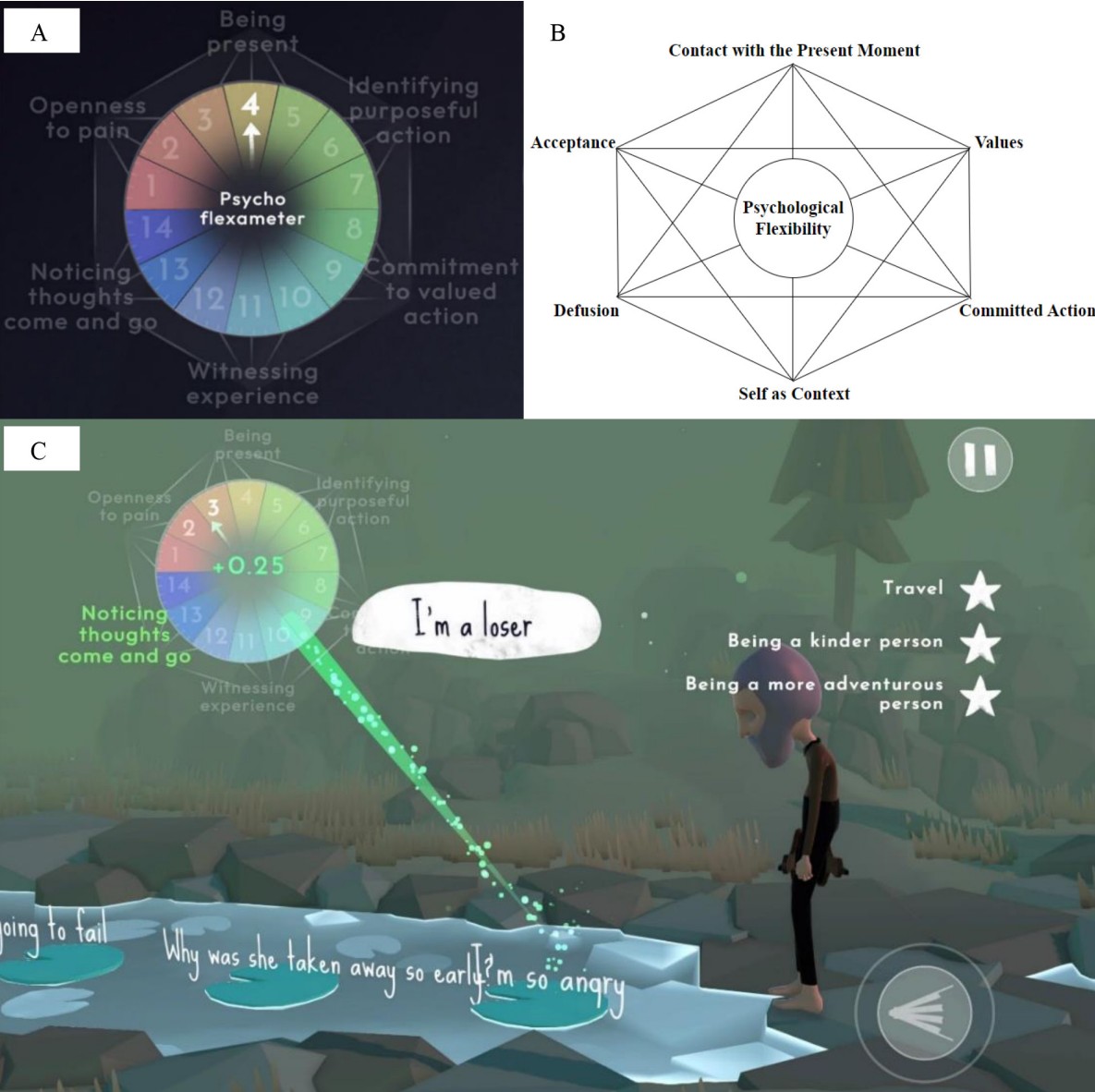

**Figure 1** (A) Screenshot from 'ACTing Minds' showing the 'Psychoflexameter'. (B) The Acceptance and Commitment Therapy Hexaflex and core processes. (C) Screenshot from 'ACTing Minds', example of 'Cognitive defusion' task. The player is required to type in their own difficult thoughts, before dragging them on to a leaf that floats downstream.

emphasises acceptance, the 'Psychoflexameter' remains visible in the corner of the screen throughout the game while players engage in other ACT-related activities. As players exhibit ACT-consistent behaviours, they earn points and gradually increase the dial on the centre of the 'Psychoflexameter', and the text reflecting the ACT processes that the players use lights up green (see online supplemental file 6). If players exhibit ACT-inconsistent behaviours, they lose points and the 'Psychoflexameter' lights up red (see online supplemental file 7).

The game starts with a text-based chapter, telling the story of a depressed individual, 'Steve', who has recently lost his wife in an accident; he is feeling depressed, isolated and lonely. The character has built a 'mind escape machine' intending to enter his own mind to destroy and suppress his unwanted painful thoughts and

memories. The player takes control of 'Steve' in chapter 2, where they see him in a state of mental distress at his home, surrounded by items that are reminders of his lost wife. At this stage, participants learn how to control the character using an onscreen directional stick and interact with the environment by touching key elements with their finger on their mobile phone or tablet.

Participants then engage with ACT content within chapters 3 and 4, which begin with the character entering his mind (via the mind escape machine), walking around and viewing painful representations of his memories (of his lost wife). Chapter 3 focuses on 'Acceptance', introducing players to a bar in the centre of the screen indicating the characters' present level of pain and discomfort, as well as the 'Psychoflexameter' dial in the corner of the screen, indicating the character's psychological flexibility. While

in the mind of the character, the player can approach memories of 'Steve' (himself) and his wife, which leads to an increase in present pain and allows the option to destroy the memories (this is intended as a metaphorical representation of thought suppression). Destroying memories decreases short-term pain and discomfort but also removes points from the 'Psychoflexameter'. If the player chooses to destroy the memories (avoidance-based strategies), the world becomes increasingly distorted, and barriers form making the chapter impossible to complete. Alternatively, if the player chooses acceptance-based strategies, they can continue the game and learn that acceptance is functionally better than avoidance (see online supplemental file 8).

Chapter 4 focuses on rewarding 'Values identification' and 'Cognitive Defusion'. The player is still in the mind of the character, where they are then asked to reflect on their values, to type them out and make them explicit (see online supplemental file 9). Following this, they complete a 'leaves on a stream' task, requiring them to type out any painful thoughts that they might have and place them on a leaf, watching them as they float downstream (see figure 1C). Both tasks reward the player by increasing their score on the 'Psychoflexameter'.

## Qualitative analysis

We used a critical realist ontological framework for our reflexive thematic analysis (RTA) as suggested by Braun and Clarke[36] which involves (after an initial familiarisation pre-coding phase) actively conducting both bottom-up (raw data driven and without a conceptual framework in mind) inductive, and top-down (ACT theory driven) deductive stages, to explore participant experience with the 'ACTing Minds' intervention. We adopted this inductive (ie, without framing the raw data through a theoretical model) first stage approach to ensure the themes developed were grounded in the raw data itself rather than potentially being imposed and biased by preconceived theories of the researcher. This is consistent with the critical realist approach which assumes that at least some of reality exists independently of our preconceived knowledge and theories, and the researcher should be actively aware of this. We then followed this with a deductive top-down ACT theory-driven second stage that then allows for a more theory-informed interpretation of the qualitative data based on ACT concepts and theory. This involved a re-examination of interview content with explicit consideration of how participant statements might relate to our research questions and ACT theory. The codes developed accordingly, transitioning from reflecting explicit semantic content to interpretations of underlying latent themes (via an ACT interpretation). This iteration of induction and deduction is important within RTA, as it allows for a more nuanced qualitative understanding of semistructured interviews, that goes beyond a purely theory-driven lens. This adopts a contextualist epistemological stance of our interpretation, recognising that both researchers' and participants' knowledge and perceptions are shaped by their subjective experiences and situational contexts.[36] This combined (critical realist and contextualist epistemological) philosophical foundation guided the application of our reflexive thematic analysis, serving as a contextualised lens for identifying and generating themes from within the interview data.

The RTA was conducted on two sets of semistructured interviews (postintervention n=29, and 3-week follow-up n=29), following the guidelines outlined by Braun and Clarke.[36] Both interview sets were analysed separately to gain an understanding of changes in participant perceptions of the intervention and relevant outcomes over time. For reporting on the acceptability of the 'ACTing Minds' intervention, findings and themes from both interviews are summarised in the primary outcomes section of the paper (see table 1).

Interview data were transcribed using Microsoft's automated audio-to-text software, which was then double checked and edited to correct for major spelling or grammatical errors. Throughout the initial data familiarisation phase, multiple points of potential analytical interest were identified. In the coding phase, several hundred codes were initially produced (both inductively and deductively), which were then clustered to make them more manageable and categorised into potential broad patterns of meaning. For the analysis of the first interview set, these included emotional experience; well-being needs, perceptions on mental-health education within the game; and participant engagement. For the second set of interviews, these were application of the game's lessons, perceptions on what was learnt, desire for growth, and sense of development.

Themes were then refined in the context of our research questions relating to how the participants experienced 'ACTing Minds', which involved a review of preliminary themes in relation to the codes, the coded data and the full dataset. We became most interested in the latent ideas underpinning statements relating to how participants used the game as a psychoeducational tool. The preliminary theme 'emotional experience' was developed, as it was interpreted from the codes that participants were using the game as a 'base for exploring and accepting difficult emotions'. Further development of the remaining themes emphasised the processes involved in participant engagement, personal therapeutic goals and feelings regarding the games' embedded learning features. Themes derived from the codes in the second interview reflect the participant outcomes since playing 'ACTing Minds'. This allowed for an exploration of specific aspects of growth, how the participants implemented insights gained from the game in the weeks that followed, and their reflections on what they had learnt through the practical application of these insights in real-life scenarios.

## Quantitative analysis

A repeated measures ANOVA was performed using IBM SPSS Statistics 29 (the most up-to-date version at the

**Table 1** Themes and sample codes taken from thematic analysis for Interviews 1 and 2

**Interview 1**
**(immediate postintervention) Inductive Codes Deductive Codes**

| Themes | Raw data (without preconceived theory)-driven inductive codes | ACT theory-driven deductive codes |
|---|---|---|
| Theme 1: desire for an engaging therapeutic experience | Need for well-being tools; surprised by effectiveness; interest in novelty; well-being development as an enjoyable practice | Psychoflexameter aids engagement; core ACT concepts useful |
| Theme 2: personal process of immersion | Empathy with story; interest in metaphor, open mind needed; personalisation aids relatability | Immersion through visual metaphors and agency; engagement through ACT therapeutic intent |
| Theme 3: game as a base for exploring and accepting difficult emotions | Anxiety while making decisions; game helped to clear head; desire for future use as a tool; mood change with game; | ACT skills applicable across emotional scenarios; long-term acceptance benefit despite difficult emotional experience; learning to be open to difficult emotions |
| Theme 4: embedded learning game dynamics pros and cons | Concepts made more sense as the game progressed; lack of instruction, but quickly learnt concepts; conflicting choices | Interpreting ACT metaphors quickly; priming of ACT-based well-being; ACT concepts clear despite confusion with game objectives |
| Theme 5: necessary learning for anyone | Desire to share with others; growing societal appeal | ACT concepts made tangible; game provided direction for growth towards ACT values; ACT concepts felt relevant |

**Interview 2**
**(3 weeks postintervention follow-up) Inductive Codes Deductive Codes**

| | | |
|---|---|---|
| Theme 6: utility in the real world | Sharing lessons with others; easier time letting go; built desire to learn more | Applying ACT lessons actively; potential real world subconscious influence of ACT lessons; increased perspective-taking in real-life events |
| Theme 7: practice facilitates psychological flexibility skills | New interest in well-being; shift in thinking; combined game and interview helpful | ACT practice encourages optimism with new ACT knowledge despite present suffering; trial and error of applying ACT-based lessons; ACT practice encourages renewed focus on values |
| Theme 8: closer alignment to an integrated self (as context), with acceptance, values, as part of who you are | Primed self-reflection; seeing the bigger picture integration about self; reduced self-judgement | Integrating present moment awareness, acceptance and values; dealing with grief through acceptance of self as I am; acceptance brings the self closer to reality; self-assurance with values; self as context identification |

ACT, acceptance and commitment therapy.

time of analysis) to compare the effects of playing the 'ACTing Minds' video game on scores taken from the questionnaires DASS-21, AAQ-II Psychological Flexibility Questionnaire, Warwick-Edinburgh Mental Well-Being Scale, Social Connectedness Score and EuroQol Five Dimensions.

Descriptive statistics were used to summarise secondary outcome measures (see table 2). Changes in scores from baseline are reported for each of the measurement time points. Partial eta squared ($\eta p2$) effect sizes were calculated for each independent variable and interpretation was informed by prior literature on the topic.[37] Values 0.14 or higher were interpreted to be a large effect, 0.06–0.14 were interpreted to be a moderate effect and 0.01–0.06 were interpreted to be a small effect.

**Procedure**

After recruitment (see 'Participant' section for recruitment), and consenting to take part in the study, participants were given a link to Qualtrics where they completed the battery of questionnaires (see 'Secondary outcome measures' section) at baseline. They were then given a link to the ACTing Minds game (see 'Intervention' section) where they completed this within approximately 1 hour. Once completed, they then immediately completed the quantitative questionnaires (see 'Secondary outcome measures' section) again on Qualtrics as an immediate postintervention. This was then followed by completing a 45 min to 1 hour one-on-one interview which asked participants about their experiences with the game (see 'Primary outcome

**Table 2** Illustrating change in intervention outcomes over time (n=29)

| | Pre-intervention baseline, mean (SD) | Postintervention, mean (SD) | Follow-up (3 weeks), mean (SD) | ηp2 | Effect size | F | Power | Full RCT sample size estimated assuming 0.8 power |
|---|---|---|---|---|---|---|---|---|
| DASS-21 depression | 14.34 (5.97) | 14.21 (5.54) | 13.90 (4.75) | 0.011 | Small | 0.31 | 0.43 | 436 |
| DASS-21 stress | 15.34 (4.55) | 14.52 (4.40) | 14.28 (4.65) | 0.108 | Medium | 3.39 | 0.999 | 46 |
| DASS-21 anxiety | 12.79 (5.19) | 12.10 (4.42) | 11.66 (4.75) | 0.096 | Medium | 2.98 | 0.999 | 52 |
| AAQ-II (psychological flexibility) | 27.34 (10.47) | 28.38 (9.86) | 28.86 (8.65) | 0.060 | Medium | 1.76 | 0.999 | 86 |
| WEMWBS | 42.07 (7.31) | 42.62 (7.91) | 42.07 (7.00) | 0.011 | Small | 0.31 | 0.33 | 436 |
| UCLA social connectedness | 64.72 (7.52) | 64.24 (8.85) | 63.90 (8.33) | 0.021 | Small | 0.60 | 0.77 | 236 |
| EQ5D mobility | 4.83 (0.38) | 4.83 (0.38) | 4.83 (0.38) | 0.000 | N/A | 0.00 | 0.05 | Negligible (no effect) |
| EQ5D self-care | 4.78 (0.58) | 4.79 (0.49) | 4.76 (0.51) | 0.000 | N/A | 0.00 | 0.05 | Negligible (no effect) |
| EQ5D usual activities | 4.14 (0.79) | 4.38 (0.68) | 4.62 (0.68) | 0.307 | Large | 12.42 | 1.00 | 16 |
| EQ5D pain/discomfort | 4.24 (0.64) | 4.45 (0.63) | 4.31 (0.81) | 0.010 | Small | 0.28 | 0.25 | 520 |
| EQ5D anxiety/ depression | 3.86 (0.99) | 4.03 (0.87) | 3.97 (0.94) | 0.018 | Small | 0.52 | 0.66 | 288 |
| EQ5D self-rated health score | 71.14 (19.01) | 72.76 (19.57) | 70.93 (20.71) | 0.000 | N/A | 0.00 | 0.05 | Negligible (no effect) |

AAQ-II, Acceptance and Action Questionnaire-second version; DASS-21, Depression Anxiety Stress Scale; EQ5D, EuroQol Five Dimension; RCT, randomised controlled trial; WEMWBS, Warwick-Edinburgh Mental Well-Being Scale.

measures' section). Participants then, after a 3-week follow-up, completed the same questionnaires again, as well as a second interview (see 'Primary outcome measures' section) that focused on real world application of the ACTing Minds intervention.

## Public and patient involvement

Key stakeholders were consulted and involved in the development of this protocol. The Patient Experience and Evaluation in Research (Patient Experience and Evaluation in Research (PEER): https://www.swansea.ac.uk/humanandhealthsciences/research-at-the-college-of-human-and-health/patientexperienceandevaluationinresearchpeergroup/) group in the College of Human and Health Sciences at Swansea University were consulted. This group represented members of the public, students and staff members, several of whom reported that they had experienced depression, anxiety or stress at some point in their lives and emphasised the need for innovative approaches to the delivery of mental health support. The feasibility design was explained to them, and they gave positive feedback about the nature of the design, intervention and outcome measures.

# RESULTS

## Primary outcome measures

### Participant recruitment and retention (feasibility)

Thirty-six participants were recruited through the initial study advertisement between 1 October and 1 December, all of which met eligibility criteria. Six participants did not show up for initial baseline measures, while one participant did not follow through with the intervention (see figure 2). Only three participants reported why they were not able to attend, where one indicated they had a hospital appointment, another had forgotten about the study date and did not reschedule, while another said they needed to reschedule without giving a reason, but then failed to book in a new date for the study. The other three participants did not report why they could not attend.

### Participant feedback (acceptability)

Acceptability measures were assessed through thematic analysis of semistructured interviews (all data are available on the Open Science Framework: https://osf.io/3wuh5/),[38] taken place immediately postintervention and at a 3-week follow-up. The interviews were analysed

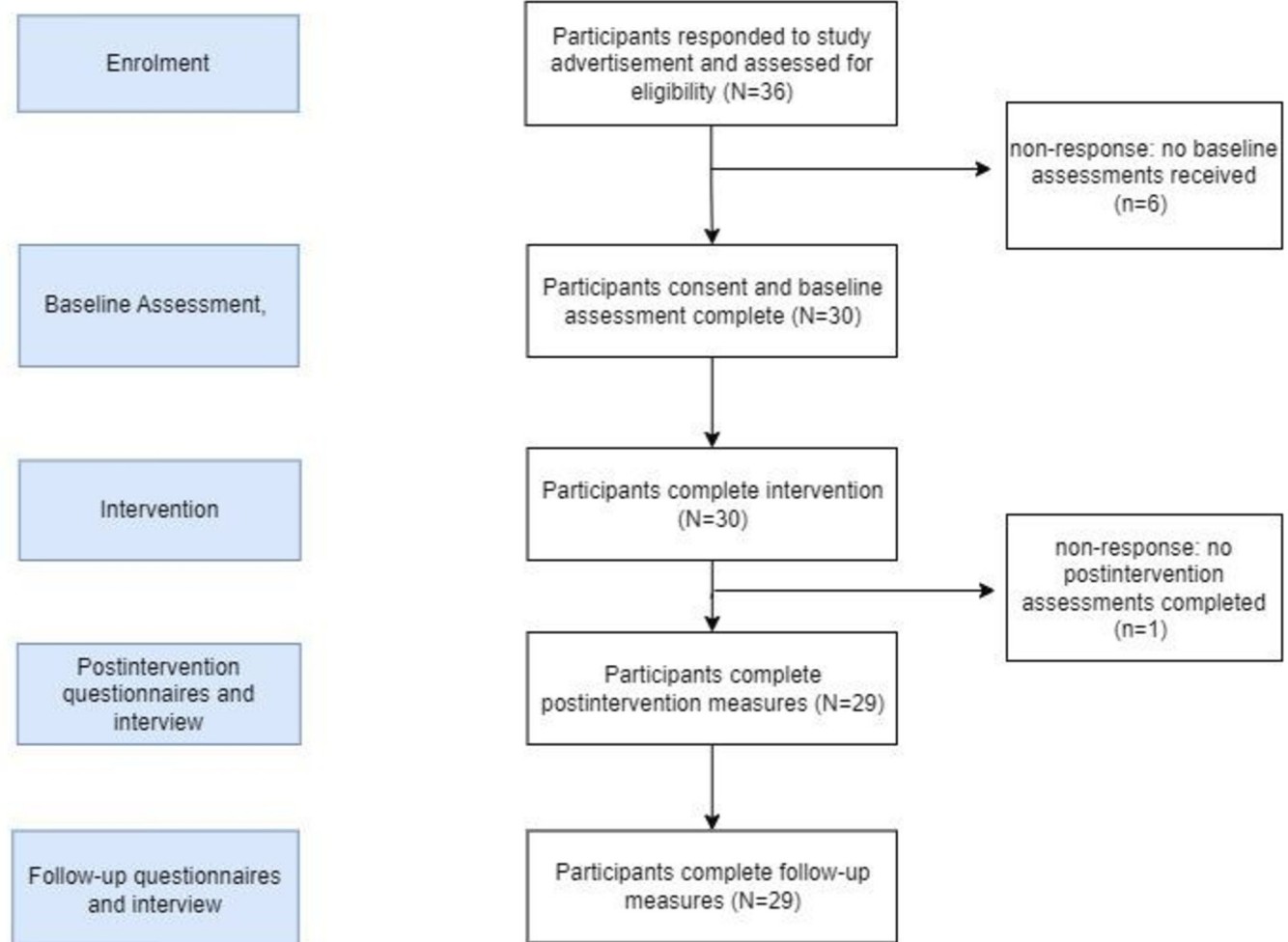

**Figure 2** Flow of participants through the study.

separately to understand participant perceptions before and after applying lessons learnt from the game to their everyday lives. The results from both have been summarised and reported together as key themes.

## THEME 1: DESIRE FOR AN ENGAGING THERAPEUTIC EXPERIENCE

Participants expressed interest in the novelty and potential utility of digital mental health applications, specifically in a gamified context. Overall, they felt that a video game platform provided a uniquely stimulating means of engaging with mental health learning. Participants suggested that psychoeducational tools like 'ACTing minds' may be able to function as an engaging alternative or an adjunct to therapy. For example:

"[…] playing a game that's based for your mental health is really good because you can actually do something therapeutical in the things that I like doing." -P06.

This quote comes from a participant who identifies as a gamer, with little experience or prior interest in mental-health learning. Their statement implies an underlying desire for an engaging, accessible therapeutic environment that he has an interest in. The intervention was also helpful for non-gamers, for example:

"I'm not one to sit down and just watch a video and then put that in practice. I'm someone who actually needs to be involved in something and interact with it, and coursework just reminds me too much of school […] with games it makes me sit, and actually interact because I'm enjoying the game at the same time. It's not just that I need to focus on my mental health, I get to play a cool game while learning about my mental health. So yeah, I like I like video games with mental health stuff." -P08

In this situation, the participant does not identify as a gamer and has substantial experience with their own mental health learning and therapy. They, like participant one, also expressed the same desire for a more engaging therapeutic experience and enjoyed the format of learning about mental health via a game. The participant expresses an aversion to learning about mental health in other more traditional ways.

## THEME 2: PERSONAL PROCESS OF IMMERSION

Responses from the semistructured interviews highlighted the individual differences between participants in their ability to immerse themselves within the content of 'ACTing Minds' and engage directly with the personal decision-making aspects of the game. Responses indicated that the video game immersion and narrative helped them visualise the ACT metaphors and thus helped their understanding of the ACT concepts within the embedded learning environment. For example:

"It plays in a very good way of dealing with visual metaphor, but also the fact that you're given the agency to do it. When you were smashing the memories and I wish I didn't have to, but I have to carry on Steve's journey and I think it's a really good way of showing these memories do hurt, and sometimes you just have to accept the fact that it's going to hurt. Move on like it's explained through the visual and narrative storytelling, and I think it's a really good way to do it." -P18

This participant refers to the visual immersion and having agency within the game as facilitating their learning of ACT concepts. The requirement for participant agency in making emotionally difficult decisions within the game appears to have facilitated the learning of acceptance.

Learning was also facilitated by the personalised nature of the game, such as selecting difficult personal thoughts in the lily pad exercise, for example:

"[…] it's been like a learning experience, and so the sort of personalised bit at the end with the lily pads and the values. I think it was quite good because then you actually were able to think about like the purpose of the game within your own situation." -P10

## THEME 3: GAME AS A BASE FOR EXPLORING AND ACCEPTING DIFFICULT EMOTIONS

Participants discussed the idea that the game provided a platform that allowed them to explore their own emotions in an immersive environment. Participants reported experiencing a variety of difficult thoughts and emotions throughout playing the game, and that the game encouraged them to observe and be open to sadness, anxiety and grief with acceptance. Participants reported feeling that after playing the game, they had learnt to observe and accept those feelings rather than to actively avoid them, and the importance of values. For example:

"At least the main bit that I got from it was to first just observe some of the negative feelings that you have, and not necessarily like reject and wrestle with them, just to sit and watch them and observe them, and accept them […]. You can perhaps do something like remind yourself of values. I like that one too." -P05

The participant clearly reflects that the game encouraged them to observe and accept their thoughts and feelings rather than suppress (or 'wrestle') them. They also mention that the game reminded them of what really matters to them.

Participants appreciated the leaves of the stream exercise as promoting acceptance of difficult thoughts, for example:

"I'd like the Lily pad thing. I think it's really nice because I myself struggle with my emotions or bad thoughts or whatever, and I stay on them, and I make

myself feel guilty about situations. So, it was really nice to accept letting go of yourself." -P06

Other participants reflected on the game encouraging acceptance and observation of difficult thoughts, for example:

"I think that section when it's like acceptance and realising that you know, even though you may not like these thoughts you're having, you still need to be aware of them and you can use them as a springboard so that was my favourite section." -P18

## THEME 4: EMBEDDED LEARNING PROS AND CONS

Data from the immediate postintervention interviews suggest a mixed response to embedded learning in 'ACTing Minds'. Many participants reported learning a variety of lessons through playing the game, most being intended in the game's design. Participants were sometimes confused with the lack of explicit objectives but were still able to understand that there were consequences to avoidant behaviour, and that acceptance of difficult emotions was rewarded. This indicated the embedded learning within the game dynamics was successfully implemented. For example:

"So when you hit them (memories), the like walls go up right, yeah? […] But I guess it was like kind of showing you that if you hit the memories, you kind of close yourself off and that's what the walls were." -P14

Another participant also suggested that they were able to learn relevant ACT concepts quickly such as not to avoid (suppress) thoughts, for example:

"I think the fact that I think it was very quickly aware that you shouldn't just destroy memories and I think that was that was done really well. Maybe too well. As in like it went on a bit."

## THEME 5: NECESSARY LEARNING FOR ANYONE

Several participants expressed that they felt what they learnt via ACTing Minds would be applicable and useful to many others in society, and not just for themselves. For example:

"I think it's a good thing that more people are learning about this kind of thing and. It kind of just leads into more research regarding and. More support out there and more help. It's generally like, I think. Maybe not my thing, but. It's not like a bad thing, it's. A good thing there's people doing it." P27.

"I think like the impact did still stick with me, like I still mentioned it quite a bit to like my parents afterwards and I mentioned it to my partner as well. They seemed interested and they like wished they could have done that too."-P18.

## THEME 6 (3-WEEK FOLLOW-UP INTERVIEW THEME): UTILITY IN THE REAL WORLD

In the interview occurring 3 weeks postintervention, participants discussed how they were able to apply the lessons that they took from 'ACTing Minds' into the real-world. While a few participants had not considered the game since playing or tried to actively apply lessons, many found that they made a conscious effort to apply ACT principles and techniques from the game into their lives. Some example quotes and descriptions are given below:

"I have noticed I've actively now, if I get like a negative thought in my brain. I try and register it and I don't hold on. But like, because sometimes before, even though I kind of subconsciously did it with some things, I just didn't (always), you know. But now if it's even something stupid like I've been lazy. I was like alright don't think about it all day. Actually go out for a walk. Go to the gym you know. Don't just keep in your head, I'm so lazy and miserable and fat. Get out to do something about it." -P04

This statement was given by a participant who in the prior immediate-postintervention interview felt that they did not have a problem with holding on to difficult thoughts. Despite this, in the weeks since playing 'ACTing Minds', the participant found that they were more conscious of how their thoughts have impacted their real life, and as a result, have been able to apply the core ACT concepts in tandem. Their statement indicates that they were able to apply defusion lessons to difficult thoughts about their self ("I'm so lazy and miserable and fat"), acceptance of their behaviours ("I've been lazy") and commitment to values by taking part in ACT-oriented activities regardless of their thoughts. Many participants noted that even if they had not explicitly tried to apply the ACT principles learnt from the game to real life, that they felt the game had influenced them subconsciously:

"I wouldn't necessarily say I've sat there and primed those thoughts [regarding ACT concepts], but in unconscious thinking, if you know like passive thought and stuff like that in my day-to-day, I've definitely had hints of some of those topics. Do you know what I mean? Like even today, I was going about my day doing my thing and you know you'll have a thought that'll throw you back to the past, and then you learn like I came to accept it." -P11

This participant found that the ACT principles were more readily available to them when confronted with real-life situations that demanded them. In these cases, participants most commonly felt that they were more accepting of difficult emotions and situations.

## THEME 7 (3-WEEK FOLLOW-UP INTERVIEW THEME): PRACTICE FACILITATES PSYCHOLOGICAL FLEXIBILITY SKILLS

A core theme present across follow-up interviews was that participants expressed how applying what they had learnt

into their day-to-day activities over 3 weeks led to an even deeper understanding of the ACT concepts they learnt. Participants reported greater engagement with values-orientated behaviours in their everyday real lives, and a greater ability to cognitively defuse (or let go) of difficult thoughts rather than engaging in avoidant behaviour. Some example quotes and descriptions are given below:

> "Well, it's like actually testing on a real situation as to just generally learning it. But then when you come into a situation, you start to understand a bit better why you do those things (referring to ACT skills) and what benefit those things have, because the situation is actually impacting on your emotions or your feelings and stuff like that, and so then you're like oh, this is why this is a good technique." -P05

In practising real-life application of ACT principles, participants were able to get a deeper sense of how and why the ACT techniques worked for them, especially during emotionally challenging situations. Through such practice, participants have noted that their personal values have become clearer to them:

> "I've learnt ways to engage with my thoughts, and like I've always tried to practice letting go of things that aren't like too meaningful, like things that won't matter in a day and all that sort of stuff. But I feel like the game has helped me also realise […] a way to really put my values down in a more straightforward manner." -P13

The participant refers to the real life practising of letting go of difficult thoughts (cognitive defusion), and values identification.

### THEME 8 (3-WEEK FOLLOW-UP INTERVIEW THEME): CLOSER ALIGNMENT TO AN INTEGRATED SELF (AS CONTEXT), WITH ACCEPTANCE, VALUES, AS PART OF WHO YOU ARE

One of the most consistent patterns across interview responses was that participants felt they had learnt more about themselves through acceptance, or that practicing the lessons taught in the game helped them align to their values. Some participants seem to have expressed that they learnt the ACT concepts in a more integrated way, where they felt that acceptance of difficult thoughts and their values was part of who they were that is, self as context. Some example quotes and descriptions are given below:

> "I actually learned a lot from this game about like these inner emotions or bad memories are not sinful. They're a part of you, and they contribute what you're going to be or the current you [self]. So yeah, that's the core lessons I guess I learned from the game." -P23

The participant reflects here about accepting their personal experiences in the present moment for what they are (indicating broader integrated acknowledgements about themselves in context). For other participants, this acceptance of personal experiences in day-to-day life has facilitated further identification of personal values:

> "Yeah, as I said, the values that's definitely helped me. Learning to just calm my thoughts for a little and think of the small but important things in life and what I appreciate that helps. It's also made me learn more about how to deal with grief […] It's helped to learn that it's OK to feel grief, it's a part of who I am, and I must accept it." -P08

### Secondary outcome measures

Quantitative results (all data are available on the Open Science Framework: https://osf.io/3wuh5/)[38] revealed a large effect size for the EQ5D usual activities score ($\eta p2=0.307$), while medium effect sizes were found for DASS-21 Stress scores ($\eta p2=0.108$), DASS-21 Anxiety scores ($\eta p2=0.096$) and AAQ-II Psychological Flexibility scores ($\eta p2=0.060$). Small effect sizes were obtained for the DASS-21 Depression scores ($\eta p2=0.011$), the WEMWBS ($\eta p2=0.011$), UCLA Social Connectedness Scale ($\eta p2=0.021$), EQ5D Pain/Discomfort ($\eta p2=0.010$) and the EQ5D Anxiety/Depression ($\eta p2=0.018$). Given this is a feasibility study that is intentionally underpowered as it has a small sample size, $p$ value significance is statistically meaningless for measuring the efficacy of any given measure. Instead, the effect sizes allow for a G-Power[39] a-priori analysis to be conducted that indicates the sample size required to detect meaningful statistical between-group differences within a future full-scale RCT. For this, the G-Power (V.3.1.9.7) indicated that when assuming a between factor (two group), repeated measures (three points in time) design with an alpha error probability of 0.05, and acceptable power of 0.8,[40] then 436 participants are required to detect a meaningful statistical difference for the smallest measure effect size (DASS-21 depression) in a future RCT. See table 2 for full details including effect sizes, power and estimated sample sizes for a future RCT given the observed effects sizes of this feasibility study.

### Convergent outcomes

Integration of results is considered a defining feature of mixed-methods research.[41] In the interest of transparency, we have included a visualisation of the research outcomes taken from our qualitative and quantitative approaches, illustrating how each outcome links back to feasibility, as well as the conclusions made following each outcome (see figure 3).

### DISCUSSION

### Statement of principal findings

The overarching aim of this research was to test the feasibility and acceptability of the 'ACTing Minds' video game as a DHI for treating depression, anxiety and stress. Participant recruitment and retention, as well as quantitative and qualitative results, demonstrate that the study

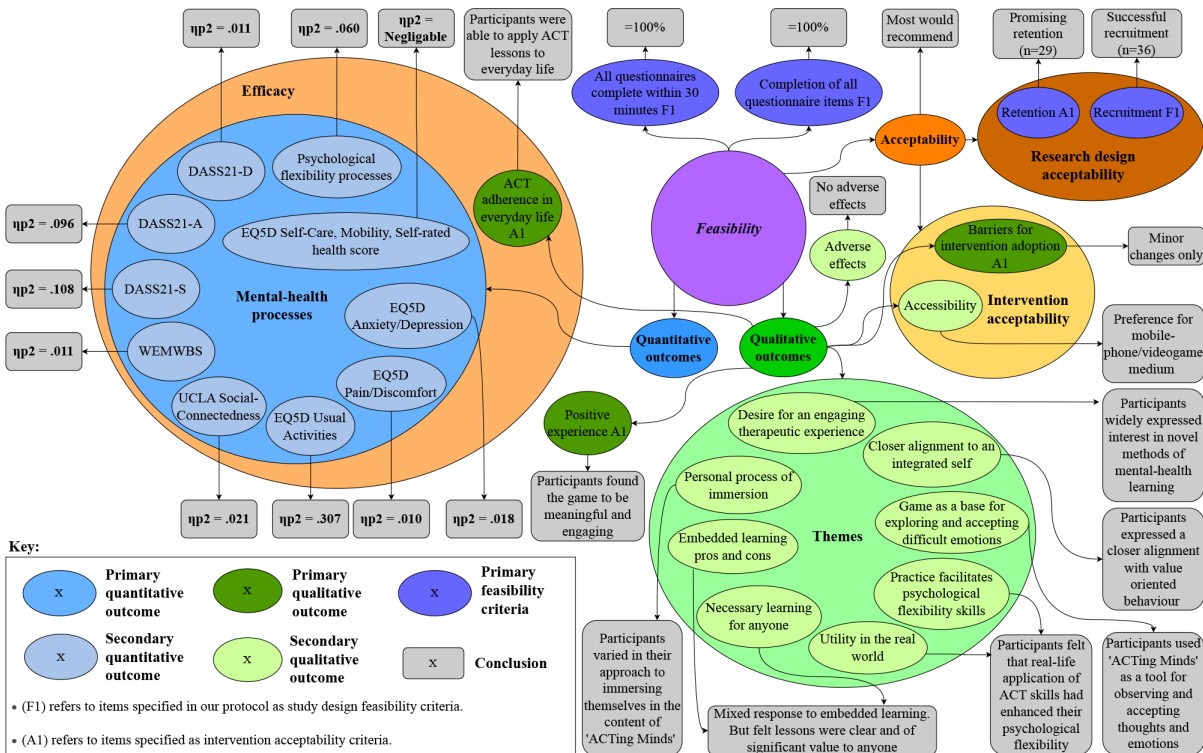

**Figure 3** Diagram visualising the integrated outcomes of 'ACTing Minds' feasibility and acceptability study.

design and video game intervention are feasible for testing in a full-scale RCT. Thematic analysis on qualitative data revealed several key findings. First, participants were successfully able to learn about core ACT principles through embedded learning, specifically acceptance, defusion from thoughts and commitment to personal values. There was also some indication that participants had learnt something about themselves in an integrated way, in the form of self as context. Participants felt that the lessons taught within the game could be applied to their daily lives and that the game was effective in priming them to consider the core ACT principles throughout the weeks following their completion of the intervention. Participants also felt that they would recommend the game to someone that they care about and that they would be interested in downloading and completing future releases of 'ACTing Minds'.

### Summary of secondary outcomes

Quantitative analysis of measures taken at baseline, immediately postintervention, and after a 3-week follow-up, revealed promising small to large effect sizes in many of the quantitative measures.[38] This included a large effect size for increasing EQ5D usual activities, medium effect sizes for reducing DASS-21 stress and anxiety, as well as increased AAQ-II psychological flexibility. Small effect sizes were observed for reducing DASS-21 depression, EQ5D pain and discomfort, EQ5D anxiety and depression and increased UCLA social connectedness and WEMWBS general well-being. There were no observed effects for EQ5D mobility, self-care and self-rated health score. Given this is a feasibility study that is intentionally underpowered as it has a small sample size, $p$ value significance is statistically meaningless for measuring the efficacy of the interventions, instead effect sizes are more informative as they express the underlying effect and are not influenced by population size. Instead, the G-Power analysis provided an estimation of participants required (436 to account for the small effect sizes) to identify meaningful significance in a full-scale future RCT.

### Comparison to existing literature

This is the first study to use a video game DHI rooted in third-wave behavioural therapy to address mental distress. 'ACTing Minds' imparts psychological skills based on ACT to promote psychological flexibility. Prior research has primarily focused on video game DHIs targeting 'illbeing' that aim to reduce symptoms of mental illness, two examples are games 'REThink' and 'Dojo'. 'REThink,' designed for a younger audience, developed the players' ability to discern functional emotions from maladaptive ones and was shown to effectively improve emotional symptoms and reduce depressive mood.[42] 'Dojo,' using biofeedback and relaxation techniques to promote emotional regulation, was shown to significantly decrease participants' anxiety and aggressive behaviour scores postintervention but exhibited no long-term effects at a 4-month follow-up.[43] Our DHI differs fundamentally from these other games due to strong theoretical underpinnings based on ACT for promoting psychological flexibility instead of a focus on reducing unwanted emotions or emotional regulation. Participants within the game were taught to observe and be open to emotional pain without judgement or any

attempt to change them, and this is a key focus within the 'ACTing Minds' intervention.

Several meta-analyses have been conducted to explore the potential efficacy of DHIs, one found that the majority of DHIs were based on cognitive behavioural therapy (CBT) and that the effect size for such interventions was small for reducing depressive symptoms compared with non-treatment controls.[44] CBT-based DHIs usually do not take the form of full video games but may include elements of gamification such as rewards, badges and progress tracking. They are typically structured programmes including online education tools, interactive exercises and self-assessment tools, which focus on challenging and modifying negative thoughts and behaviours. A meta-analysis including 34 RCTs (17 of which were CBT-based) found that CBT-based DHIs yielded a medium effect size for reducing symptoms of depression and anxiety.[45] However, a meta-analysis of 117 CBT-based applications revealed that only 12 of them provided support aligned with the evidence-based tenets of CBT.[46] This finding suggests that the observed effectiveness in earlier studies could potentially stem from participants' interaction with a DHI (perhaps as a form of distraction) rather than their proficient implementation of CBT principles. CBT-DHI programmes often require consistent use, and high attrition rates have limited their efficacy in research.[47] One meta-analysis author suggested that DHIs may need to be complemented by existing mental health support.[44] However, our study challenges this notion. In less than 1 hour of playing 'ACTing Minds', participants fully explored the game, retained ACT knowledge and discussed its positive real-world application in interviews conducted 3 weeks postintervention.

### Strengths and limitations
Our study is the first to explore the feasibility and acceptability of a novel video game DHI based on ACT. A core strength of this research was the utilisation of a mixed-methods approach. By incorporating thematic analysis of semistructured interviews as well as quantitative analysis of questionnaire data, we were able to gain a comprehensive understanding of participant experience using 'ACTing Minds'. Collecting quantitative data at three separate time points, and interviews conducted at two separate time points meant that we were able to examine the gradual processes of change and identify patterns of improvement consistent with the ACT model. We gained valuable input from participants in terms of suggestions for improving the intervention which will aid in the further development of 'ACTing Minds' to optimise effectiveness and user engagement. The results from the interviews also indicated that 'ACTing Minds' has broad appeal as a video game even to those outside of clinical populations.

It is also important to acknowledge the limitations of our study. First, there was no control group, though this is intentional as this is a feasibility study, it is only when we conduct a full RCT that we will have an adequate comparator group to determine whether the intervention is clinically useful at promoting psychological flexibility and reducing depression, anxiety and stress. There was a lack of in-game data logging of specific ACT tasks completed. For example, we did not collect data on whether participants actually entered text about the difficult thoughts that they were experiencing such as in the leaves on a steam exercise, though all participants completed the game. The reliance on self-report measures potentially allowed for biased responses, including psychosociological measures such as HRV would strengthen the study outcomes. It is also possible that some of the questionnaires used in this study were insufficient for capturing the target measurements. One study used exploratory factor analysis to investigate the extent to which the AAQ-II Psychological Flexibility Questionnaire can discriminate between experiential avoidance and psychological flexibility. The researchers found that AAQ-II items were more strongly related to items measuring distress than items measuring acceptance.[48] In line with this, the AAQ-II has been criticised as being too simple a measure for psychological flexibility.[49] In a future RCT, we may adopt another measure for measuring psychological flexibility such as The Personalised Psychological Flexibility Index which may be a more valid measure of psychological flexibility.[50] Finally, though thematic analysis can be highly useful for identifying shared meaning and variation among the themes, and bridging subjectivity and theoretical structure, it also has limitations. The contextualist epistemology used in this approach acknowledges the researcher has an active role in shaping the outcomes and can be biased by their own knowledge and experience. The subjective nature of the themes means that there can be variation in the interpretation of the data between different thematic researchers. So, though this approach can be useful, these limitations also need to be acknowledged.

### Clinical implications and directions for future research
The ACT-based video game DHI used in this study is a low-cost, engaging and easy-to-disseminate means of supporting those experiencing mental health difficulties. The present study highlights the clinical implications of 'ACTing Minds', including its potential therapeutic value, user engagement and accessibility. However, further research is warranted to establish long-term effects, explore specific populations, conduct comparative studies, investigate underlying processes and address any ethical considerations that may arise. Critically, a full RCT is now needed, in which participants are compared quantitatively with a control group, incorporating physiological well-being measures such as HRV, as well as research-validated questionnaires regarding mental health (ie, depression, stress, anxiety, psychological flexibility, social connectedness and well-being). By pursuing these future research directions, we can leverage the potential of ACT-based video games such as 'ACTing Minds' to enhance

patient care, improve outcomes and expand the reach of interventions in an increasingly digital era.

## CONCLUSION

The results of this study demonstrate that 'ACTing Minds' is feasible to implement in a full-scale RCT. Both the intervention and study were well received by participants, thematic analysis of semistructured interviews indicated that a single playthrough of the game was sufficient for teaching several core principles of ACT, namely 'Acceptance', 'Values Identification' and 'Cognitive Defusion' to participants, and priming them to implement the lessons in their day-to-day lives. There was some evidence that participants also integrated their learning about themselves as self as context, which is interesting. Quantitative results indicate that playing 'ACTing Minds' is associated with decreases in depression, anxiety and stress, as well as increases in psychological flexibility, social connectedness and well-being. However, these effects will need to be further explored in an adequately powered RCT to understand the potential clinical implications, therapeutic value, user engagement and accessibility of an ACT-based video game DHI.

**Acknowledgements** We would like to thank Miricle Tea Studios Ltd and Mikoshi Ltd for the production and release of the ACTing Minds video game. We would also like to thank Prof. Louise McHugh (University College Dublin), Dr. Alison Stapleton (Dublin Business School) and Dr. Sarah Cassidy (Smithfield Clinic) for several helpful comments on various aspects of the ACTing Minds intervention and methodology.

**Contributors** DJE and AHK designed the original protocol, whilst TCG updated and revised the protocol design. TCG wrote the first draft of the paper and conducted all of the quantitative and qualitative results. DJE and AHK provided substantial revisions on all drafts and advised TCG throughout the development of this manuscript. DJE designed and developed the game development. TCG acts as the guarantor for this study as first author, responsible for the overall content.

**Funding** Funding for the development of this game came from European development funds via the commercial entity of Swansea University called AgorIP (Reference: 229-0256-0046) awarded to DJE.

**Competing interests** The game ACTing Minds was developed using European development funds via the commercial entity of Swansea University (AgorIP) and awarded to DJE with the intention to develop this game for commercial purposes (as a game app for the Apple and Google Play stores). DJE was involved in the design of the protocol but did not recruit participants, collect any data, and did not conduct the analysis on the data. TCG and AHK have no involvement in any commercial aspects of the game.

**Patient and public involvement** Patients and/or the public were involved in the design, or conduct, or reporting, or dissemination plans of this research. Refer to the Methods section for further details.

**Patient consent for publication** Consent obtained directly from patient(s).

**Ethics approval** This study involves human participants and was approved by School of Psychology sub-committee Swansea University 2022-5630-4834. Participants gave informed consent to participate in the study before taking part.

**Provenance and peer review** Not commissioned; externally peer reviewed.

**Data availability statement** Data are available in a public, open access repository. Data are available on the open science framework which is linked to within the paper.

**ORCID iDs**
Andrew H Kemp http://orcid.org/0000-0003-1146-3791
Darren J Edwards http://orcid.org/0000-0002-2143-1198

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
