## [Reviewer comments · BMJ Open]

ARTICLE DETAILS

TITLE (PROVISIONAL)	Mixed-methods feasibility outcomes for a novel ACT-based videogame 'ACTing Minds' to support mental health
AUTHORS	Gordon, Tom; Kemp, Andrew; Edwards, Darren

VERSION 1 – REVIEW

REVIEWER	Sarah Cassidy Smithfield Clinic
REVIEW RETURNED	07-Dec-2023

GENERAL COMMENTS	BMJ Review Suggestions for edits Abstract, page 2- Clause— “Using a standardised battery of questionnaires” needs to be changed to full sentence Page 4, -Strengths and Limitations off study needs to be changed to of study. Page 6, end of first full paragraph should be consider, rather than considers P14- Laughing at the character being called Steve. Is that a coincidence or a solid incidence of good old fashioned Brown Nosing? Page 15, under Qualitative Analysis- I think some of the language here is unnecessarily verbose. I would try to try to simplify this a little to make it more parsimonious. There's nothing here that's untrue. It just feels inaccessible. Page 16, suggest changing “fell easier” to “fell more easily” Page 17, suggest tidying up Table 1 as this looks a little sloppy/uneven. Suggest referencing Interviews 1 and 2 with “Interviews” starting with a capital letter on the title of this table. Page 17, under Quantitative Analysis, all first letters of titles of scales should be in caps (e.g., Psychological Flexibility Questionnaire, Social Connectedness Score) as opposed to just the first letter of the entire scale (Psychological flexibility questionnaire and Social connectedness score) as these are the formal names of these scales and this needs to be consistent with how you've referenced the other scales in this section and throughout the article. Page 18, under Participant Recruitment and Retention, suggest that
---

"1" participant should be instead listed as "one" participant in line with most guidelines for referencing of numbers for publications, unless perhaps this journal or these authors has/have a different stylistic preference.

Page 20, towards the end of the page- Very good points here but I suggest breaking up some of these sentences as the length of them is interfering with clarity. No more than two clauses per sentence is generally a good rule of thumb.

Page 21, I feel that there is a little bit of speculation around what the participants may or may not have been getting out of different aspects of their game play. I would remove or completely re-phrase the first full paragraph on page 21 starting with the words, "This participants' statement puts significance on the more personalised sections of the game...." I'm not sure the authors are incorrect but this feels highly speculative. Possibly if the authors could state what they mean a little more clearly here, it wouldn't feel so speculative. I would also avoid use of polarised language or even attaching a valence that is necessarily positive or negative to the game players' thoughts which we can't necessarily know. I tend to say that thoughts may be "difficult" or "unwanted" which is more consistent with ACT approaches. Using the language of negative versus positive thinking is seen more in the CBT literature than the ACT literature which may use words like "difficult" or "unwanted" thoughts or experiences but tends not to use negative versus positive per se.

Page 21. Theme 3 – The language in this section is not at all ACT consistent so this section needs to be completely re-worked. I'm not sure what this sentence means, "Participants regularly invoked the idea...". The point of ACT is not to help people to feel "calm" so that they can engage with emotionally intensive emotions. Thoughts and emotions (even those ones that are intense, I'm not sure I'd say intensive here??) happen in life anyway, no matter whether people feel calm or don't feel calm. Similarly, I'd again urge caution around the use of the word "negative" in describing emotions because if this game is set up as a game to help us to feel calm so that we can accept negative emotions then it is missing the point of ACT altogether. There are a vast array of situations in life that may never ever be calm so if we can only ever cope with life when we manage to bring about calm then we are going to be in big trouble when real storms hit, no matter who invokes what or how they manage to do the invoking.

Page 23- This quotation illustrates.....

Again, I think the authors may be assuming a little too much about what these quotations illustrate and possibly over-using the quotations at this point in the article. I like the use of some quotations for thematic analysis etc but it's very difficult to speculate what individual quotations are actually illustrating in any kind of empirical way. Once again, I'm urging caution with the use of "negative emotions" as it may look like the goal of the game is to help people to face negative emotions, but really thoughts are just thoughts and people have all kinds of emotions. Avoidance is the issue, not the valence of any particular emotion per se. Overly engaging with positive emotions or positively valenced thoughts items is also psychologically problematic. Fusion is a problem no matter whether the thought or emotion has a positive or negative valence so the issue is not the valence. The issue is the fusion.

Page 24. Second full paragraph. Incomplete sentence starting with the word "But"

Page 25. Second full paragraph. ACTing Minds written as ACTting Minds.

Page 27. Under Theme 7. Similar to my comments on the use of the word "Negative", I think it's also not wise to frame "positive" thinking as the goal here and certainly, this would not be consistent with the ACT framework. Perhaps that was still what people reported and if that's the case then that's the case, but I'm always cautious about setting that up as a goal because then you're setting people up for a fall when they have a bad day and they can't muster up anything positive. In fact, the research shows that telling people to "Just Think Positive" on a day like that makes them feel worse. This is in line with the research on thought suppression. So, non-judgemental awareness of what your thoughts are is perhaps a better way of looking at this. If we start framing "Positive Thinking" as our goal, we're stepping into pop psychology spaces and not science and that's truly worrisome. Sometimes people do genuinely feel more positive as an outcome of being more mindfully aware, but that really shouldn't be the goal. It feels like that is what this is saying here.

P29- "The growth experienced is intangible"- That is not what I would take from this person's quotation at all. Again, I'd really question the speculation on what the participant was thinking, feeling or saying here. I'm not sure if it is the word-choices to describe the participant's experiences that are just not hitting the mark here or just that this one word just doesn't seem like it's the right one for what this particular quote seems to be expressing. I understand that Thematic Analysis is a very difficult task (and one that I don't usually undertake for that very reason!), but I would actually completely avoid Thematic Analysis if you can't find words that parsimoniously explain what the themes are. To me, this particular quote never said anything at all about being "intangible" so I'm finding myself annoyed every time I read an analysis of a quote and I'm wondering if Chat GPT analysed it because some of these descriptions of what the themes are seem to be so far removed from what the quote seems to be saying that it isn't a thematic analysis at all! It's just words strung together for the sake of stringing them together. It would be better to just remove this section if it's too difficult to summarise what the participant was saying or if we don't know what the theme was. I truly don't think this participant said one thing about the tangible or intangible nature of their experiences. If they did, it wasn't in this quote. Sorry to be so pedantic. It's not my intention but Thematic Analysis is genuinely difficult and I think that's why a lot of scientists possibly just don't do it.

Page 33—There is clearly much better understanding here around what the ACT approach is, but this seems inconsistent with several points earlier in the paper which seem to be suggesting that negative thoughts were a problem and getting rid of them was a desired outcome, or thinking less negatively was more desirable. So I think I'd go through this section here and make sure that what you've very clearly stated here is crystal clear everywhere else in the paper—particularly in the thematic analysis sections where everything feels much looser and less like it fits with the coherent structure of the rest of the paper.

Page 35, end of first paragraph, list out full name of Social Connectedness measure

Page 35- under Clinical Implications....

Second line in this section—I'd re-word "Those who suffer mental distress" to something along the lines of "those experiencing mental health difficulties". Mental distress seems an antiquated terminology.

Overall, I think this is a wonderful piece of work and I'm really genuinely excited to see it being done. I think some tidying up of the style of the writing here would make this overall piece a lot clearer. There are a few pieces throughout the work here that feel ACT inconsistent to me and that needs to be clarified so that the whole piece is cohesive and coherent with the stated goals of teaching ACT through this video game. The goals are really clear and consistent on page 33 but, as stated, there are a few places within the body of the article where I was unsure if perhaps the main author was suggesting that getting rid of negative thoughts was actually the goal of the game or teaching the player to be calm so that they would be able to deal with negative thoughts was the goal of the game. So, I'd really like to see a move away from even setting up a polarising of "calm versus chaotic", "good versus bad" thoughts and emotions because that is just more polarisation of everything. That's the whole problem with parents who don't want their kids playing any videogames, or people who can't any carbs, good and evil, depression and anxiety. Polarisation of anything is generally problematic. Moderation of lots of stuff is usually fine. Avoidance of living a valued life due to fusion with rigid rules—those are things that are problematic. Thoughts are just thoughts. Context matters. That's the whole point. It's not actually about if someone is calm or not calm, but about whether or not they can engage with a life that matters alongside of whatever thoughts they might be having, even if those thoughts might be unwanted or indeed deemed to be unpleasant at that time.

I'm not sure Thematic Analysis was the way to go with analysing this data set as there were a lot of pieces here that felt very speculative. As I stated above, I don't know if it was just that the word choices used to describe what the game players'/participants were saying didn't capture it for me or perhaps the word choices were just not the correct ones but this felt really "loose" and I think there were way too many quotes used here that just lacked the structure that they needed for me to be sure they were actually necessary for this paper. In future, I'd either not use Thematic Analysis at all or have much clearer guidelines for how the themes should be described in the write up. This part felt way too disjointed to me. I like what these pieces can add to an analysis more generally, but in this case, the Thematic Analysis did not add to the article for me. The Thematic Analysis was, imho, very shaky. Please know that this is not my area of expertise, but I think this specific section needs a very careful reviewing to ensure that what the authors are suggesting the themes are, are actually what the participants' seem to be saying. My suspicion is that they aren't. (Sorry).

Again, though, really genuinely happy to see work like this being done and looking forward to seeing it in print because I feel that this is very important work to get out to the world. This type of work could reach populations that might never ever make it to clinic or

	counselling settings. The data are looking really promising and I imagine that as gamification further iterations of this project come out, this will only get better. Looking forward to seeing the final edited version of this piece. Well done to the authors for this important work. Thank you for the opportunity to review it. -Sarah Cassidy
--	--

REVIEWER	Alison Stapleton Dublin Business School
REVIEW RETURNED	10-Dec-2023

GENERAL COMMENTS	An exciting project with promising outcomes that can meaningfully inform future RCTs. I commend the team for their hard work and success. > In the introduction, I recommend rephrasing "by practicing emotional control" to instead focus on one's relationships with emotions. On page six, when explaining how ACT might improve on previous approaches to wellbeing, I think "emotional regulation" (ER) should be omitted - as this paper stands, ER could be understood as being incompatible with ACT/the third-wave when in fact acceptance is arguably compatible with ER. This issue also appears later in the Discussion ("a focus on regulating") and should be amended, in my opinion. > When discussing Scholten et al.'s Rayman 2 and Dojo study, recommend explicitly naming the duration of their follow-ups (i.e., no superiority in the short OR long term). > Some typing errors throughout, e.g., "ACT is a third wave behavioural therapy, which prioritise", "videogame called 'ACTing Mind'", spelling out the ACT acronym in the Discussion. > I think this line should be removed or rephrased: "does not necessitate formal clinical training or accreditation in order to be applied effectively". I think that the intention of this line could be misunderstood. If retaining this line, I recommend instead citing some of the recent ACT self-help literature. > "ACT aims to decrease suffering" does not align with the previous framing of ACT as distinct from symptom alleviation and must be amended. At the same time, caution is needed here since the authors are using distress as a secondary outcome. > What pre-existing theoretical frameworks were used for the deductive components of the Qual analysis? > Some phrasing in the Qual sections is inconsistent with the epistemological approach: e.g., "From there many of the remaining codes fell easier into place," and "painted a picture" - recommend rephrasing to better align with the active stance evident elsewhere in this manuscript. > Are subthemes missing from Table 1? Would be great to see themes > subthemes > Codes > Sample quotes > Did the six participants not returning for initial baseline measures give reasons? These could be good to report given the feasibility study aims. > In the results, themes could be presented much more concisely - substantial repetition of the quotes in their corresponding theme summary. Recommend prioritizing integration in reporting. > Advise against including stats numbers in the Discussion. > In summarizing the secondary outcome measures in the Discussion, I think the authors should address significance and
---

	statistical power here and later in the limitations section. > In the Discussion, for consistency, I recommend sticking to "less than one hour" rather than 40 minutes for the duration of ACTing Minds. > Recommend referring to the literature on the AAQ-II as not distinct from emotional distress when discussing limitations.
--	--

VERSION 1 – AUTHOR RESPONSE

Reviewer 1

Reviewer 1> Happy to review this manuscript and to see this important work being done. Have made suggestions for revisions below. Many kind thanks for your consideration.

Authors> Many thanks for reviewing this paper! We really appreciate it, and you have provided some really helpful feedback. Thank you!

Reviewer 1> Abstract, page 2- Clause— “Using a standardised battery of questionnaires” needs to be changed to full sentence

Authors> Thanks, and this is now done.

Reviewer 1> Page 4, -Strengths and Limitations off study needs to be changed to of study.

Authors> Thanks for spotting a typo, this is now corrected.

Reviewer 1> Page 6, end of first full paragraph should be consider, rather than considers

Authors> This is now corrected.

Reviewer 1> P14- Laughing at the character being called Steve. Is that a coincidence or a solid incidence of good old fashioned Brown Nosing?

Authors> The first name is somewhat of a homage, though the character is actually entirely novel and fictional :-). We are glad though this made you smile :-)

Reviewer 1> Page 15, under Qualitative Analysis- I think some of the language here is unnecessarily verbose. I would try to try to simplify this a little to make it more parsimonious. There’s nothing here that’s untrue. It just feels inaccessible.

Authors> Thanks this has now been simplified, and more to the point.

Reviewer 1> Page 16, suggest changing “fell easier” to “fell more easily”

Authors> Thank you for this suggestion, we have now made the change.

Reviewer 1> Page 17, suggest tidying up Table 1 as this looks a little sloppy/uneven. Suggest referencing Interviews 1 and 2 with “Interviews” starting with a capital letter on the title of this table.

Authors> Thanks, we have made several changes to Table 1 following suggestions from you and reviewer 2. This has involved a reformatting of the table to include some sample codes from the thematic analysis as well as some general tidying as you have suggested.

Reviewer 1> Page 17, under Quantitative Analysis, all first letters of titles of scales should be in caps (e.g., Psychological Flexibility Questionnaire, Social Connectedness Score) as opposed to just the first letter of the entire scale (Psychological flexibility questionnaire and Social connectedness score) as these are the formal names of these scales and this needs to be consistent with how you've referenced the other scales in this section and throughout the article.

Authors> Thanks, this is now done.

Reviewer 1> Page 18, under Participant Recruitment and Retention, suggest that "1" participant should be instead listed as "one" participant in line with most guidelines for referencing of numbers for publications, unless perhaps this journal or these authors has/have a different stylistic preference.

Authors> Thank you for this suggestion, we have now changed the in-text numbers into words where the numbers are smaller than 10 throughout the manuscript, in line with Vancouver guidelines that this journal subscribes to.

Reviewer 1> Page 20, towards the end of the page- Very good points here but I suggest breaking up some of these sentences as the length of them is interfering with clarity. No more than two clauses per sentence is generally a good rule of thumb.

Authors> Thank you, we have now corrected this.

Reviewer 1> Page 21, I feel that there is a little bit of speculation around what the participants may or may not have been getting out of different aspects of their game play. I would remove or completely re-phrase the first full paragraph on page 21 starting with the words, "This participants' statement puts significance on the more personalised sections of the game...." I'm not sure the authors are incorrect but this feels highly speculative. Possibly if the authors could state what they mean a little more clearly here, it wouldn't feel so speculative. I would also avoid use of polarised language or even attaching a valence that is necessarily positive or negative to the game players' thoughts which we can't necessarily know. I tend to say that thoughts may be "difficult" or "unwanted" which is more consistent with ACT approaches. Using the language of negative versus positive thinking is seen more in the CBT literature than the ACT literature which may use words like "difficult" or "unwanted" thoughts or experiences but tends not to use negative versus positive per se.

Authors> Thanks, these are really helpful points. We have reworked this section and reduced overly speculative statements. We have also replaced the terms such as "positive" and "negative" in favor of words such as "difficult" or "unwanted". It is also important to note that we do use an inductive and deductive approach in our thematic analysis, it is only in the deductive stage do we apply a theory driven (ACT terminology) lens, so we do allow for some terms to be inductively included in our coding that maybe less ACT consistent or familiar to an ACT audience (as the lay players being interviewed are largely unfamiliar with the ACT terms), this is to prevent any bias as much as we can. We have made this approach clearer in this section. However, we have included some of the more ACT-consistent codes from the deduction process, so the section is much more ACT-consistent now.

Reviewer 1> Page 21. Theme 3 – The language in this section is not at all ACT consistent so this section needs to be completely re-worked. I'm not sure what this sentence means, "Participants regularly invoked the idea...". The point of ACT is not to help people to feel "calm" so that they can engage with emotionally intensive emotions. Thoughts and emotions (even those ones that are intense, I'm not sure I'd say intensive here??) happen in life anyway, no matter whether people feel calm or don't feel calm. Similarly, I'd again urge caution around the use of the word "negative" in describing emotions because if this game is set up as a game to help us to feel calm so that we can accept negative emotions then it is missing the point of ACT altogether. There are a vast array of

situations in life that may never ever be calm so if we can only ever cope with life when we manage to bring about calm then we are going to be in big trouble when real storms hit, no matter who invokes what or how they manage to do the invoking

Authors> Thanks, again, this is a really helpful suggestion. We have now used the more neutral words such as “difficult” or “unwanted”. We have also now included examples where there is clear evidence of the participants expressing that they were leaning into difficult emotions whilst learning to engage with their values and create a space of openness and acceptance. As with the previous point, we do allow for some codes that are also less ACT familiar if they are being generated by the participants and are inductively generated in the thematic analysis so as to avoid any bias from ourselves as ACT researchers. We have now also included some of the more ACT-consistent codes from the deduction process, so the section is much more ACT-consistent now.

Reviewer 1> Page 23- This quotation illustrates..... Again, I think the authors may be assuming a little too much about what these quotations illustrate and possibly over-using the quotations at this point in the article. I like the use of some quotations for thematic analysis etc but it's very difficult to speculate what individual quotations are actually illustrating in any kind of empirical way. Once again, I'm urging caution with the use of “negative emotions” as it may look like the goal of the game is to help people to face negative emotions, but really thoughts are just thoughts and people have all kinds of emotions. Avoidance is the issue, not the valence of any particular emotion per se. Overly engaging with positive emotions or positively valenced thoughts items is also psychologically problematic. Fusion is a problem no matter whether the thought or emotion has a positive or negative valence so the issue is not the valence. The issue is the fusion.

Authors> Thanks, we have again now used more neutral terms such as “difficult” or “unwanted” thoughts rather than “positive” or negative”, and have now toned down speculation, now focusing on more clear examples, or acceptance, openness and cognitive defusion.

Reviewer 1> Page 24. Second full paragraph. Incomplete sentence starting with the word “But”

Authors> Thanks, this has now been changed as requested now starting with “However, they...”

Reviewer 1> Page 25. Second full paragraph. ACTing Minds written as ACTting Minds.

Authors> Thanks for spotting this, this has now been changed.

Reviewer 1> Page 27. Under Theme 7. Similar to my comments on the use of the word “Negative”, I think it's also not wise to frame “positive” thinking as the goal here and certainly, this would not be consistent with the ACT framework. Perhaps that was still what people reported and if that's the case then that's the case, but I'm always cautious about setting that up as a goal because then you're setting people up for a fall when they have a bad day and they can't muster up anything positive. In fact, the research shows that telling people to “Just Think Positive” on a day like that makes them feel worse. This is in line with the research on thought suppression. So, non-judgemental awareness of what your thoughts are is perhaps a better way of looking at this. If we start framing “Positive Thinking” as our goal, we're stepping into pop psychology spaces and not science and that's truly worrisome. Sometimes people do genuinely feel more positive as an outcome of being more mindfully aware, but that really shouldn't be the goal. It feels like that is what this is saying here.

Authors> Thanks, and we agree with you. As with previous points we have now used more neutral terms such as “difficult” or “unwanted” thoughts rather than “positive” or negative. Though these are the terms that participants use, we have now opted for more a theory driven lensing ACT lensing as the lay community are likely to use more surface level concepts that they are more familiar with. In

Table 1 we give examples of the ACT-theory deductive codes (i.e., lensing what the participants report through ACT theory concepts) as well as the more inductive and raw coding of the surface level language participants use. We have also included in the limitations section, that this is a problem of thematic analysis generally, that if we rely too heavily on the raw lay statements of participants this is likely to be a surface level analysis, and therefore some theory-driven analysis (such as ACT concepts) are required.

Reviewer 1> P29- “The growth experienced is intangible”- That is not what I would take from this person’s quotation at all. Again, I’d really question the speculation on what the participant was thinking, feeling or saying here. I’m not sure if it is the word-choices to describe the participant’s experiences that are just not hitting the mark here or just that this one word just doesn’t seem like it’s the right one for what this particular quote seems to be expressing. I understand that Thematic Analysis is a very difficult task (and one that I don’t usually undertake for that very reason!), but I would actually completely avoid Thematic Analysis if you can’t find words that parsimoniously explain what the themes are. To me, this particular quote never said anything at all about being “intangible” so I’m finding myself annoyed every time I read an analysis of a quote and I’m wondering if Chat GPT analysed it because some of these descriptions of what the themes are seem to be so far removed from what the quote seems to be saying that it isn’t a thematic analysis at all! It’s just words strung together for the sake of stringing them together. It would be better to just remove this section if it’s too difficult to summarise what the participant was saying or if we don’t know what the theme was. I truly don’t think this participant said one thing about the tangible or intangible nature of their experiences. If they did, it wasn’t in this quote. Sorry to be so pedantic. It’s not my intention but Thematic Analysis is genuinely difficult and I think that’s why a lot of scientists possibly just don’t do it.

Authors> Thanks we have worked on these thematic sections a bit, so that they more clearly reflect what is being reported. We have used clearer and obvious instances of openness, acceptance, cognitive defusion, etc., throughout these sections. We also agree that thematic analysis is very speculative, and open to interpretation and have included this as a limitation in the limitations section. We hope though, now these are clearer, and more ACT-consistent throughout.

Reviewer 1> Page 33—There is clearly much better understanding here around what the ACT approach is, but this seems inconsistent with several points earlier in the paper which seem to be suggesting that negative thoughts were a problem and getting rid of them was a desired outcome, or thinking less negatively was more desirable. So I think I’d go through this section here and make sure that what you’ve very clearly stated here is crystal clear everywhere else in the paper—particularly in the thematic analysis sections where everything feels much looser and less like it fits with the coherent structure of the rest of the paper.

Authors> Thanks, we now have made it clear throughout the paper that promoting acceptance, openness to painful thoughts, and values orientation is the focus of the intervention, rather than reduction of negative experience. We have now expressed this more thoroughly through the themes generated, so this should now be more consistent throughout.

Reviewer 1> Page 35, end of first paragraph, list out full name of Social Connectedness measure

Authors> Thanks, this is now changed to “The Social Connectedness Scale”.

Reviewer 1> Page 35- under Clinical Implications.... Second line in this section—I’d re-word “Those who suffer mental distress” to something along the lines of “those experiencing mental health difficulties”. Mental distress seems an antiquated terminology.

Authors> Thanks and done.

Reviewer 1> Overall, I think this is a wonderful piece of work and I'm really genuinely excited to see it. Looking forward to seeing the final edited version of this piece. Well done to the authors for this important work. Thank you for the opportunity to review it.

Authors> Thank you again Sarah for your extremely thorough and helpful feedback, and the time you spent reviewing this paper, which means a lot to us. This has no doubt strengthened this manuscript greatly and we really appreciate it. :-)

Reviewer: 2

Reviewer 2> In the introduction, I recommend rephrasing "by practicing emotional control" to instead focus on one's relationships with emotions. On page six, when explaining how ACT might improve on previous approaches to wellbeing, I think "emotional regulation" (ER) should be omitted - as this paper stands, ER could be understood as being incompatible with ACT/the third-wave when in fact acceptance is arguably compatible with ER. This issue also appears later in the Discussion ("a focus on regulating") and should be amended, in my opinion.

Authors> Thanks for this, we have made this change in the introduction and discussion.

Reviewer 2> When discussing Scholten et al.'s Rayman 2 and Dojo study, recommend explicitly naming the duration of their follow-ups (i.e., no superiority in the short OR long term).

Authors> Thanks, the follow-up duration has now been explicitly mentioned.

Reviewer 2> Some typing errors throughout, e.g., "ACT is a third wave behavioural therapy, which prioritise", "videogame called 'ACTing Mind'", spelling out the ACT acronym in the Discussion.

Authors> Thanks, these typos have now been corrected throughout.

Reviewer 2> I think this line should be removed or rephrased: "does not necessitate formal clinical training or accreditation in order to be applied effectively". I think that the intention of this line could be misunderstood. If retaining this line, I recommend instead citing some of the recent ACT self-help literature.

Authors> Many thanks for pointing this out, we have now removed this line.

Reviewer 2> "ACT aims to decrease suffering" does not align with the previous framing of ACT as distinct from symptom alleviation and must be amended. At the same time, caution is needed here since the authors are using distress as a secondary outcome.

Authors> Thanks, we have changed this to say "ACT aims to promote psychological flexibility". We have now also mentioned that though in practice ACT does not seek to reduce depression, anxiety etc., these are commonly used indicators for clinical efficacy within research practice.

Reviewer 2> What pre-existing theoretical frameworks were used for the deductive components of the Qual analysis?

Authors> Inductive components of the thematic analysis are non-theory driven, whilst deductive components are theory driven through an ACT theoretical lens. Table 1 now include coding examples for both inductive and deductive components.

Reviewer 2> Some phrasing in the Qual sections is inconsistent with the epistemological approach: e.g., "From there many of the remaining codes fell easier into place," and "painted a picture" - recommend rephrasing to better align with the active stance evident elsewhere in this manuscript.

Authors> We have now highlighted a reflective active stance in both the inductive and deductive approaches.

Reviewer 2> Are subthemes missing from Table 1? Would be great to see themes > subthemes > Codes > Sample quotes

Authors> Thanks, this is a good point, we have now included both the inductive (non-theory driven) and deductive (ACT theory driven codes). Both of these methods are important in RTA.

Reviewer 2> Did the six participants not returning for initial baseline measures give reasons? These could be good to report given the feasibility study aims.

Authors> We did not formally ask participants why they did not show up, but three did contact us saying they were unable to attend for several reasons. One reported a hospital appointment, another said they had forgot about the experiment and apologised, whilst another said they would need to reschedule, but then did not follow up with a rescheduling date. We have added this information into the manuscript.

Reviewer 2> In the results, themes could be presented much more concisely - substantial repetition of the quotes in their corresponding theme summary. Recommend prioritizing integration in reporting.

Authors> Thanks, we have reworked the themes, removed repetition, and made the section overall more concise.

Reviewer 2> Advise against including stats numbers in the Discussion.

Authors> Thanks, these are now removed.

Reviewer 2> In summarizing the secondary outcome measures in the Discussion, I think the authors should address significance and statistical power here and later in the limitations section.

Authors> Thanks. We have now included power calculations, which have given us an estimate of the sample size needed in a full RCT, given the effect sizes we have and assuming power of 0.8 as a minimum requirement. The p value significance levels of the findings are less relevant as this is a small feasibility sample size so is (intentionally) underpowered. We have now made these points clear in the paper.

Reviewer 2> In the Discussion, for consistency, I recommend sticking to "less than one hour" rather than 40 minutes for the duration of ACTing Minds.

Authors> Thanks, this is now done.

Reviewer 2> Recommend referring to the literature on the AAQ-II as not distinct from emotional distress when discussing limitations.

Authors> Thanks, good point, this reference and discussion is now added.

Authors> Many thanks to the reviewer for these really thorough and helpful suggestions!

VERSION 2 – REVIEW

REVIEWER	Sarah Cassidy Smithfield Clinic
REVIEW RETURNED	24-Feb-2024

GENERAL COMMENTS	This paper has been significantly improved since it's original submission. I'm excited to see this and feel that this type of work is much needed in the world. This creative work will likely have broader reach than many other ways of reaching populations that might not otherwise ever access therapeutic spaces. By the way--I was originally viewing a document from a different place in this portal and there appeared to be lots of different sections still highlighted in yellow-- so I'm not 100% sure why I could even see that version, but there were still some minor text revisions that appeared to still be necessary in that version. However, that may have actually been a draft rather than the very final edition- but just to be aware that this is visible to the reviewers for some reason. Well done to the team. Delighted that this kind of work is being done!
---

REVIEWER	Alison Stapleton Dublin Business School
REVIEW RETURNED	20-Feb-2024

GENERAL COMMENTS	Thank you for your thoughtful consideration of the reviewer comments. I do recommend amending the reference to "emergent" when discussing themes to instead be "generated". There are several debates around the appropriateness of "emergent"-type language given the active role a researcher adopts in RTA. Congrats!
--

VERSION 2 – AUTHOR RESPONSE

Reviewer 1> This paper has been significantly improved since it's original submission. I'm excited to see this and feel that this type of work is much needed in the world. This creative work will likely have broader reach than many other ways of reaching populations that might not otherwise ever access therapeutic spaces. By the way--I was originally viewing a document from a different place in this portal and there appeared to be lots of different sections still highlighted in yellow-- so I'm not 100% sure why I could even see that version, but there were still some minor text revisions that appeared to still be necessary in that version. However, that may have actually been a draft rather than the very final edition- but just to be aware that this is visible to the reviewers for some reason. Well done to the team. Delighted that this kind of work is being done!

Authors> Thanks, you so much again for your helpful review! We have made some additional very minor typo corrections. You will be happy to know we will be developing further ACTing Mind games in the future so will keep you up-to-date with developments! Thanks again so much for your helpful feedback! :-)

Reviewer 2> Thank you for your thoughtful consideration of the reviewer comments. I do recommend amending the reference to "emergent" when discussing themes to instead be "generated". There are several debates around the appropriateness of "emergent"-type language given the active role a researcher adopts in RTA. Congrats!

Authors> Thanks, we have now changed “emergent” to “generated” as requested. Thanks so much for your time in providing such encouraging and helpful feedback!